# Identification of Trace Components in Sauce-Flavor Baijiu by High-Resolution Mass Spectrometry

**DOI:** 10.3390/molecules28031273

**Published:** 2023-01-28

**Authors:** Jinfeng Ge, Yulin Qi, Wenrui Yao, Daohe Yuan, Qiaozhuan Hu, Chao Ma, Dietrich A. Volmer, Cong-Qiang Liu

**Affiliations:** 1Institute of Surface-Earth System Science, School of Earth System Science, Tianjin University, Tianjin 300072, China; 2Tianjin Key Laboratory of Earth Critical Zone Science and Sustainable Development in Bohai Rim, Tianjin University, Tianjin 300072, China; 3Department of Chemistry, Humboldt-Universität zu Berlin, 12489 Berlin, Germany

**Keywords:** sauce-flavor Baijiu, trace components, ESI FT-ICR MS, molecular features

## Abstract

Sauce-flavor Baijiu is one of the most complex and typical types of traditional Chinese liquor, whose trace components have an important impact on its taste and quality. Fourier transform ion cyclotron resonance mass spectrometry (FT-ICR MS) is one of the most favorable analytical tools to reveal trace molecular components in complex samples. This study analyzed the chemical diversity of several representative sauce-flavor Baijiu using the combination of electrospray ionization (ESI) and FT-ICR MS. The results showed that ESI+ and ESI− exhibited different chemical features characteristic of trace components. Overall, sauce-flavor Baijiu was dominated by CHO class compounds, and the main specific compound types were aliphatic, highly unsaturated with low oxygen, and peptide-like compounds. The mass spectral parameters resolved by FT-ICR MS of several well-known brands were relatively similar, whereas the greatest variability was observed from an internally supplied brand. This study provides a new perspective on the mass spectrometry characteristics of trace components of sauce-flavor Baijiu and offers a theoretical foundation for further optimization of the gradients in Baijiu.

## 1. Introduction

Chinese liquor (Baijiu), vodka, whiskey, brandy, rum and gin are known as the six major distilled liquors in the world [1]. With a history of more than 6000 years, its unique aroma and taste have played a special role in Chinese traditional culture, as well as occupying an important position in the development of the national economy [2,3]. According to relevant statistics, almost 7.16 million kiloliters of liquor were produced in 2021, with sales revenue of USD 90.495 billion [4]. Compared with the production of other distilled liquors, the brewing of Baijiu is a unique and complex process [5]. The conventional production and processing of Baijiu involve five main steps, which are raw material grinding, daqu making, alcoholic fermentation, distillation, and aging [6]. Depending on the raw materials and production process, Baijiu is classified into 12 flavors [7], with strong, light, sauce and rice aromas being widely considered to be the four major types of Baijiu according to their nationwide production and sales volume [8].

Complex raw materials, special processes, and co-fermentation of multiple strains make the composition of Baijiu extremely complicated. In most Chinese liquors, water and ethanol account for over 98%, whereas aromatic substances such as acids, esters, alcohols and aldehydes account for less than 2%. The latter fraction is made up of complex organic compounds that essentially consist of carbon, hydrogen, oxygen and other heteroatoms such as nitrogen and sulfur. These so-called volatile organic compounds essentially account for the detailed classification of different liquor types [9]. Therefore, it is particularly important to investigate specific Baijiu-related compounds. In recent years, a series of studies have been conducted to explore the functional components of traditional Chinese liquor. Most of the research methods utilized spectroscopy, chromatography and mass spectrometry (MS). For instance, Fan and Qian analyzed the aroma compounds in Baijiu (including Yanghe Daqu, Wuliangye and Jiannanchun) using a gas chromatography–olfactory method and concluded that lipids were the main contributors to the aroma of liquor [10,11]. Zhu et al. [12] proposed an integrated two-dimensional gas chromatography (GC)/time-of-flight mass spectrometry (TOF-MS) method to characterize volatile compounds in Chinese liquors based on the study of flavor substances in Maotai liquor, and a total of 528 components were identified in liquor samples, including organic acids, alcohols, esters, ketones, aldehydes, acetals, lactones and nitrogen- and sulfur-containing compounds. In addition, an analytical method based on fluorescence spectroscopy was developed to distinguish liquor products manufactured from six different distilleries in East Asia and North America [13]. Subsequent studies have combined multivariate statistics and fluorescence spectroscopy to differentiate the ages of Chinese liquors [3,14]. For example, a comprehensive fingerprinting classification of flavored Chinese Baijiu using ultra-high-performance liquid chromatography-Orbitrap MS and food histology identified 29 compounds, 15 organic acids, 8 esters, and some carbonyl compounds which were associated with aging [15].

On the other hand, scholars overseas have performed a series of studies on vodka, whisky and champagne, using MS. State-of-the-art MS coupled with high mass resolving power and mass accuracy has become an ideal tool for studying complex mixtures in wine without traditional sample separation and purification steps [16,17,18,19]. For example, Rubert et al. performed a preliminary identification of markers using accurate mass measurements from MS and subsequent tandem MS spectra, with the help of different software packages and online libraries, and specific flavonol glucosides and polyphenols were identified as wine markers depending on the grape variety [20]. In 2017, Kew et al. used isotopic fine structure analysis for the first time to confirm the assignment of high-molecular-weight CHOS class species in Scotch whisky, and utilized a variety of visualization techniques and multivariate analysis to identify the key compounds distinguish various types of whisky [21]. For sure, aroma is an important attribute affecting the perceived quality of Baijiu, which has a unique collection of volatile chemicals [22]. Further identification of complex mixtures usually requires high-resolution MS, which can resolve multiple peaks with the same nominal m/z [23]. Fourier transform ion cyclotron resonance mass spectrometry (FT-ICR MS) with ultra-high mass resolution has been widely utilized to explore the signatures of complex samples at the molecular level, and to reveal elemental formulas (CcHhOoNnSsPp) of tens of thousands of compounds simultaneously [24,25,26]. Negative mode electrospray ionization (ESI−), a soft MS ionization method that produces intact gas-phase ions directly from solution, has been widely used for the analysis of natural organic matter (NOM) with abundant hydroxyl, phenol and carboxyl functional groups [27,28]. However, considering that sauce-flavor Baijiu, including its aroma, fame and value, is the premier of the Chinese liquor industry group, with the characteristic components of its flavor being dominated by compounds such as organic aldehydes, ketones, esters, amino and nitrogen compounds, which are actually better ionized via positive mode electrospray ionization (ESI+) [29,30,31], these main aroma components have not yet been adequately well determined [32].

In this study, ESI (+/−) FT-ICR MS was used to analyze the corresponding trace components of the representative sauce-flavor Baijiu thoroughly, including several representative brands: Moutai, Langjiu, Xijiu, and an internal supply sauce-flavor Baijiu, so as to obtain the overall molecular profiles of the sauce-flavor Baijiu, and the chemical diversity of different Baijiu together with the possible influencing factors. Overall, this work offers a theoretical reference for the preservation and improvement of Chinese liquor.

## 2. Results and Discussion

### 2.1. Comparison of Trace Components Obtained from Mass Spectrometry

ESI is a soft ionization technique [33,34] that allows a larger range of polar compounds to be ionized [33,35]. In general, ESI+ ionizes CHO compounds such as esters and ketones; CHON compounds containing alkali nitrogen such as amines; and CHOS compounds such as sulfoxide and sulfone. ESI− ionizes acidic CHO compounds with carboxyl and hydroxyl groups; neutral nitrogenous CHON compounds; and CHOS- and CHNOS-containing organosulfates.

Mass spectrometric analysis in combination with both positive and negative ion modes enables a more comprehensive molecular profile and contributes to a better understanding of the composition of liquor components [36]. Table 1 shows the MS characteristics of the assigned molecular formulas from both ESI+ and ESI−. A principal component analysis (PCA) based on these parameters can offer a better overview of the features and variations among different Baijiu samples. The visualization results are demonstrated in Figure 1. It can be seen that significant differences exist in the component characteristics detected by the two modes due to different ionization mechanisms. PC1 is able to explain 55.1% of the variations, wherein ESI− is mainly distributed on the negative axis of the PC1 and ESI+ is mainly distributed on the positive axis. Specifically, it can be observed that the average molecular weight (m/z_wa_) of the detected Baijiu components in the positive ion mode ranges from 355.6 to 395.3, which is higher than the corresponding range of 291.0 to 387.9 in the negative ion mode. The number of carbon atoms (C_wa_) and that of heteroatoms (N_wa_, S_wa_) in ESI+ are also significantly higher than those obtained in the negative ion mode. Comparatively, a higher abundance of components was obtained in the negative ion mode, with a nominal oxidation state of carbon (NOSC) between −0.7 and −0.1, higher than that in the positive ion mode with a range from −0.8 to −0.4. This is probably because that ESI− preferably detects a large number of compounds with carboxyl and sulfate groups that have a high degree of carbon oxidation states themselves. Additionally, the oxygen content of the detected components in ESI− is also significantly higher, which is closely related to its inherent ionization properties. It can be easily seen that the component parameters acquired from the three Maotai (MT) samples are more consistent with those from Langjiu (LJ) and Xijiu (XJ), whereas the component parameters of an internal supply brand (IS) have the most specificity.

Analysis of the main subcategories revealed that ESI+ analysis of the trace components of the six Baijiu samples showed similar relative abundances of CHO and CHNOS class species, accounting for 35.2–48.0% and 31.5–42.6% of the mass spectral contribution, respectively. On the other hand, the mass spectra acquired by ESI− were dominated by CHO and CHOS class species, accounting for 56.4–68.1% and 11.3–32.7% of the total detected ion intensity, respectively (Figure 2). It is noteworthy that the molecules revealed by ESI+ provided a contribution of 2.9–4.4% to the total ion response from the CHN class of trace components of Baijiu. Regarding the CHO class, the relative abundance of its oxygen content and oxygen number showed a normal distribution with a concentrated distribution between 3 and 9. IS exhibited a different behavior, with a more widely discrete distribution of oxygen numbers. Overall, the oxygen number of CHO class compounds detected in the negative mode was relatively higher (Figure 3).

The positive and negative ion modes revealed a wider range of compounds (Figure 4 and Table 2), which further confirmed the complementary results from different pathways, and this will help to better understand the characteristics of Baijiu more comprehensively. In general, these trace components were mainly concentrated in the upper left region in the van Krevelen diagrams. The positive ion mode is more suitable for the detection of components with a low O/C ratio (0–0.2), whereas the negative ion mode is better at detecting the components with a high O/C ratio (over 0.2). With the same ionization source, sauce-flavor Baijiu contains lower H/C and lower O/C molecular compounds and lacks lower H/C and higher O/C molecular compounds when compared with the components identified from distilled whisky [19,37].

### 2.2. The General Characteristics of Sauce-Flavor Baijiu

A typical feature of sauce-flavor Baijiu is what is described as a persistent, prominent sauce aroma that is sweet, elegant, and delicate with a long-lasting aftertaste [29]. The main raw materials of sauce-flavor Baijiu are sorghum and millet. Considering the high abundance of unique compounds in trace components in Baijiu revealed by both ESI+ and ESI−, which ranged from 59.9% to 85.1% (Table 2), the combination of the two ion modes was highly recommended for the MS characterization of sauce-flavor Baijiu samples.

According to the MS results, a total of 5934, 4413, 4126, 2845, 3080 and 4557 molecular formulas were detected in samples MT-08, MT-15, MT-18, LJ, IS and XJ, respectively, whose molecular diversity is similar to that of other distilled spirits such as Whisky [17,19,21]. Based on the classification criteria of compounds by Merder et al. [38], the molecular formulas obtained from the Baijiu samples were classified into six chemical categories, which are aliphatic, condensed aromatics, highly unsaturated with high oxygen, highly unsaturated with low oxygen, peptide-like compounds, and polyphenols; the results are plotted in Figure 5. The main types of compounds in sauce-flavor Baijiu were aliphatics, highly unsaturated with low oxygen, and peptide-like compounds. Aliphatics occurred in the highest abundance among trace components in MT-08, MT-15, MT-18, LJ and XJ, ranging from 31.0% to 43.8%. The largest content in IS was peptide-like compounds, with an abundance of 19.5%. The second most abundant portions of MT-08 and XJ were highly unsaturated with low oxygen compounds with relative abundances of 27.7% and 23.2%, respectively. The second most abundant compound in IS was aliphatics at 18.3%. The rest were peptide-like compounds, with abundances ranging between 25.2% and 29.0%.

As plotted in Figure 6, the compounds of trace components in sauce-flavor Baijiu were dominated by CHO class compounds, with the relative abundance ranging from 43.0% to 56.8%, followed by CHNOS, CHOS and CHNO classes. The distribution for other classes was 19.4–32.0% for CHNOS, 10.2–15.4% for CHOS, and 7.3–15.6% for CHNO compounds, respectively. Considering the highest relative abundance of the CHO class, which represents the major portion of the samples, the relationship between carbon number and DBE for the CHO class compounds was plotted in Figure 7. Such diagrams were used to obtain further compositional characteristics of the sauce-flavor Baijiu and the corresponding variations that existed among them. The carbon numbers of different samples were mainly distributed from 6 to 40. MT, LJ and XJ had similar characteristics, with oxygen numbers mainly ranging from 1 to 8 and DBE distribution between 0 to 8 and 13 to 16. Lower DBE values have lower carbon numbers, whereas higher values have higher carbon numbers. IS had a wide range of oxygen content, from 1 to 18, and DBE values ranged from 0 to 6. The oxygen content is higher for low DBE values, whereas the higher the carbon number the higher the oxygen content is. To better illustrate the major contents in Baijiu, the top 10 compounds with the highest abundances in each of the six samples were listed in Table 3. For MT, the highest abundance was identified to be CHNOS class compounds, and in addition, CHO class compounds were more abundant in other liquors.

### 2.3. Factor Analysis of Trace Components Characteristics of Sauce-Flavor Baijiu

Although all of the samples originated from sauce-flavor Baijiu, different types of liquor and different years of the same type of liquor also showed variations. Overall, IS is the most distinctive of the sauce-flavor Baijiu. For the same type of MT, the more aged products were assigned a greater number of molecular formulas. In terms of specific compounds, the relative abundance of highly unsaturated with low oxygen and highly unsaturated with high oxygen was significantly higher in MT-08 than in MT-15 and MT-18. However, the relative abundances of peptide-like compounds and condensed aromatics in MT-08 were significantly lower than those of MT-15 and MT-18. Among them, the amount of CHO class compounds in liquor originally produced in 2008 was 49.9% higher than that produced in 2015 and 2018, whereas its CHNOS class content accounted for 25.9%, which is relatively lower (Figure 6). As can be seen in Table 1, the oxygen content of MT-08 (O_wa_ 4.3, 5.8) was higher than that of MT-15 (O_wa_ 4.1, 5.2) and MT-18 (O_wa_ 4.1, 5.2), regardless of ESI+ or ESI−. NOSC reflects the oxidation state of carbon [39], and by calculating the relative abundance of different carbon oxidation state ranges, it was found that NOSC was concentrated from −0.93 to −0.44, and meanwhile, IS had a relatively high NOSC from −0.05 to 0.54 overall, but with a significantly lower percentage of NOSC over 1.51 compared to other liquors. According to the NOSC calculated here, it can be observed that the relative abundance distribution of the oxidation state of carbon is more consistent for MT (MT-08, MT-15 and MT-18), followed by LJ and XJ, and the largest difference remains for IS. In addition, a slight increase in low NOSC (−0.93, −0.44) and a slight decrease in high NOSC (−0.05, 0.54) were also observed from MT-08 and MT-15/MT-18 (Figure 8). Consequently, the above data offer solid evidence that aging is one of the crucial factors for the oxidation of the compounds in Baijiu.

## 3. Materials and Methods

### 3.1. Samples

Four representative types of sauce-flavor Baijiu brands with a total of six samples were selected for mass spectrometric analysis. They were Maotai manufactured in the year 2008, 2015, and 2018 (MT-08, MT-15, MT-18), Langjiu (LJ), Xijiu (XJ), and an internal supply brand (IS). All Baijiu samples were filtered with 0.45 μm pillow filters and analyzed by FT-ICR MS in ESI+ and ESI− directly.

### 3.2. Mass Spectrometry

Mass spectra were recorded using a 7T solariX 2XR FT-ICR MS instrument (Bruker Daltonik GmbH, Bremen, Germany). Baijiu samples were infused at a flow rate of 200 μL/h. For each spectrum, 256 scans were accumulated per run, and the mass range was recorded from 150 to 1000 Da. Spectra were further processed in DataAnalysis 5.0 (Bruker) and Composer 1.5.6 (Sierra Analytics) with internal calibration based on known species of homologous series throughout the entire m/z range. In order to minimize the interferences from the noise, only spectral peaks with signal-to-noise ratios above 6 were considered. An error value of ±0.5 ppm was used to assign the correct molecular formulas.

### 3.3. Data Analysis

The relative peak intensities, normalized to all molecular peak intensities per sample, were used to semi-quantitatively assess the respective molecular formulas [25,40]. The DBE values for assigned molecular formulas were calculated according to the definitions from Koch et al. [41]: DBE = 1 + 1/2(2C-H+N+P). The NOSC was calculated according to Riedel et al. [42]: NOSC = 4 − [(4C+H−3N−2O−2S)/C]. Principle component analyses (PCA) were carried out using weight-averaged values of mass spectral parameters. Statistical analysis and graphical plotting were performed in Excel 2021 and R 3.6.1.

## 4. Conclusions

Baijiu is a great invention of ancient China. From ancient times to the present, liquor has been closely connected to the life and culture of the Chinese people. Sauce-flavor Baijiu is among the four well-known traditional flavors of Baijiu and plays an irreplaceable role in Chinese liquor culture. Here, high-resolution FT-ICR MS was utilized to analyze the chemical diversity of representative samples from sauce-flavor Baijiu, and in total, over 2845 compounds were detected. The results show that a combined analysis of both ESI+ and ESI− is effective for a more comprehensive characterization of the trace components in Baijiu. ESI+ reveals substances with higher saturation and heteroatom content, whereas ESI− detects substances with higher oxygen content as well as the oxidation state of carbon. If the CHNOS class content of trace components are investigated in Baijiu, the ESI+ mode should be considered. The relative abundance distribution of CHO class compounds among trace components in sauce-flavor Baijiu ranged from 43.0% to 56.8%, CHNOS from 19.4% to 32.0%, CHOS from 10.2% to 15.4%, and CHNO from 7.3% to 15.6%. Specific compound types were dominated by aliphatic, highly unsaturated with low oxygen, and peptide-like compounds. The highest abundance of trace components in MT, LJ and XJ were aliphatic, with abundances between 31.0% and 43.8%. The highest content in IS was peptide-like compounds. Overall, the mass spectral characteristics of MT, LJ and XJ were more comparable, whereas IS showed greater variation. For MT, higher oxygen content and oxidation compounds were observed from aged MT-08 when compared to MT-15 and MT-18, implying that age is an important factor that influences the composition of sauce-flavor Baijiu. Standard ESI is one of several mass spectrometry ionization methods utilized most for wine analysis, and further combinations of other ionization methods are needed to better characterize the specific trace components in Baijiu.

## Figures and Tables

**Figure 1 molecules-28-01273-f001:**
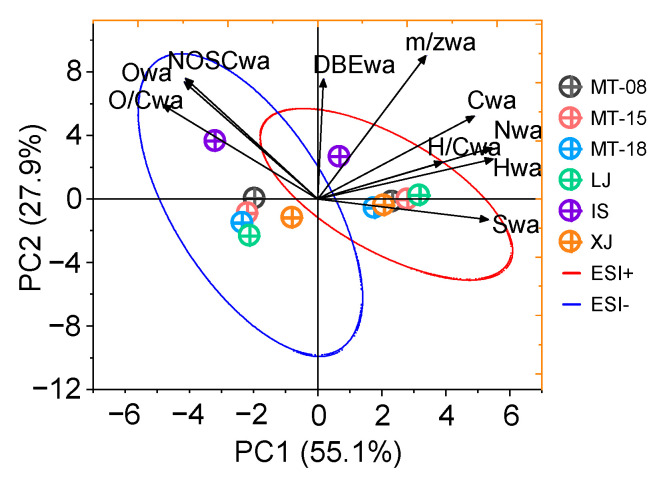
Principle component analysis of intensity-weighted average values is listed in Table 1 for the Baijiu samples measured by ESI+ and ESI−. The red confidence ellipse is the parameter measured by the ESI+ mode and the blue confidence ellipse is the parameter measured by ESI− mode.

**Figure 2 molecules-28-01273-f002:**
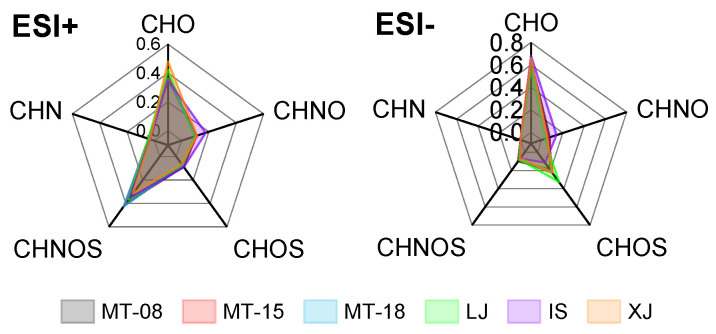
Radar diagrams show the contribution of the main subcategories measured by ESI+ (**left**) and ESI− (**right**).

**Figure 3 molecules-28-01273-f003:**
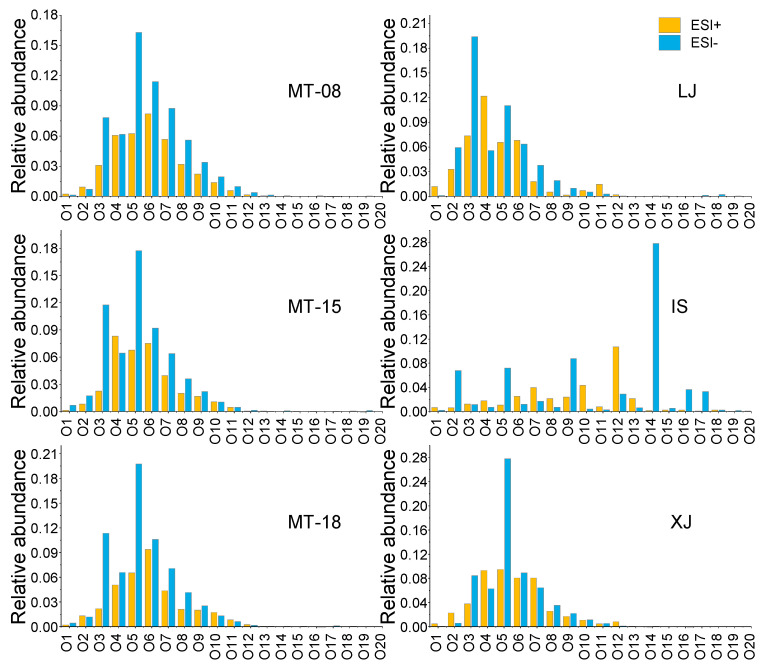
Relative abundance of Ox class compounds detected by ESI+ and ESI− in different sauce-flavor Baijiu. Yellow represents the results measured by ESI+ mode and blue represents the results measured by ESI− mode.

**Figure 4 molecules-28-01273-f004:**
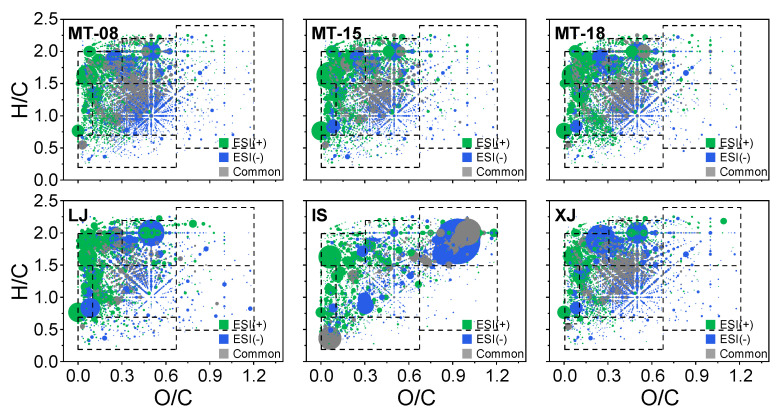
Van Krevelen diagrams demonstrating the compounds identified from different sauce-flavor Baijiu samples. The size of the dots represents the abundance of the assigned molecular formulas. The colors represent the unique and common molecular formulas detected by ESI+ and ESI−. The corresponding molecular formulas are listed in detail in the Appendix A.

**Figure 5 molecules-28-01273-f005:**
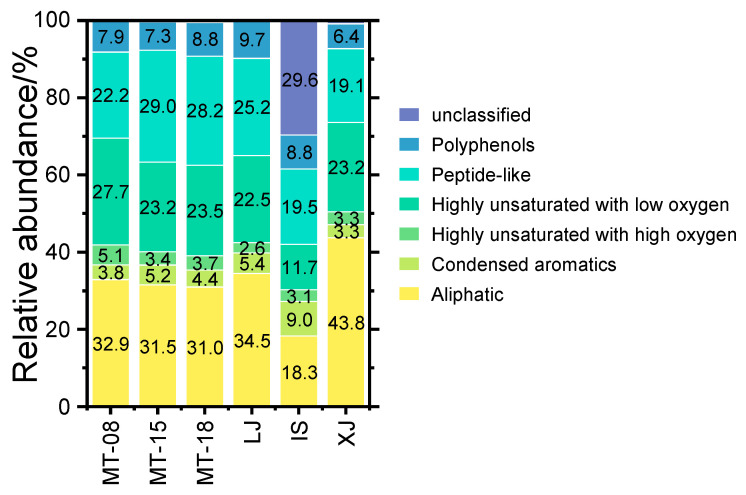
Bar diagrams showing the contribution of major compound categories in the different Baijiu samples revealed by the combination of ESI+ and ESI− mass spectrometry analysis.

**Figure 6 molecules-28-01273-f006:**
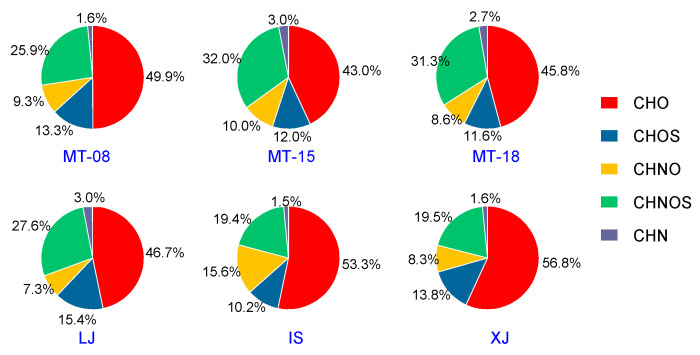
Pie diagrams showing the contribution of the main compound classes revealed by the combination of ESI+ and ESI− mass spectrometry analysis. Specific brands of Baijiu are indicated beneath the graphics.

**Figure 7 molecules-28-01273-f007:**
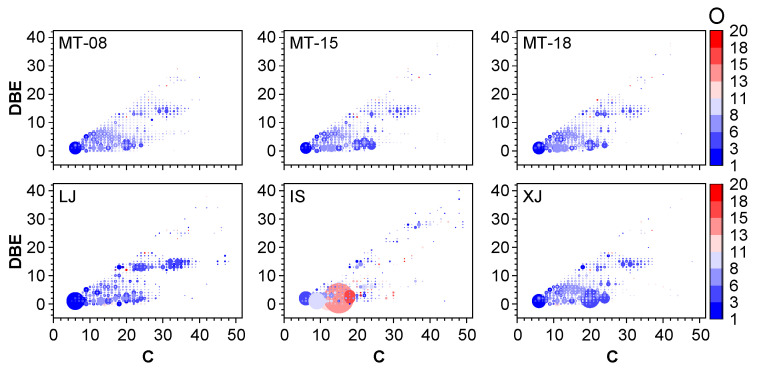
DBE vs carbon number plots for CHO class compounds in each sauce-flavor Baijiu. The size of the dots represents the abundance of the assigned molecular formulas. The colors represent the oxygen contents.

**Figure 8 molecules-28-01273-f008:**
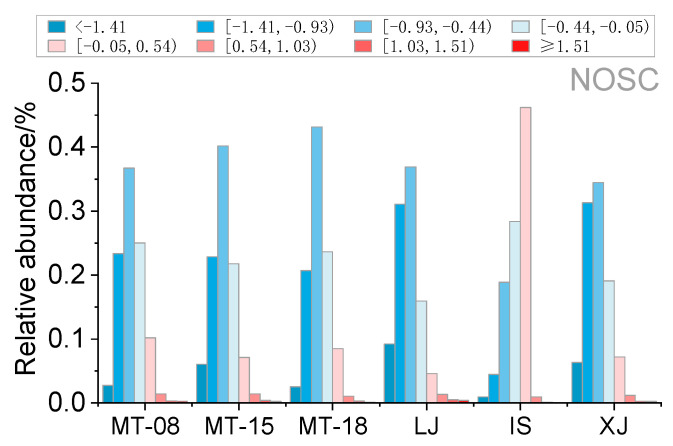
The relative abundance of different NOSC ranges in each sauce-flavor Baijiu sample. The colors represent the different NOSC ranges.

**Table 1 molecules-28-01273-t001:** Molecular parameters of trace components in Baijiu derived from ESI FT-ICR MS *.

Type		No. of Assigned Formulas	m/z_wa_	C_wa_	H_wa_	O_wa_	N_wa_	S_wa_	H/C_wa_	O/C_wa_	DBE_wa_	NOSC_wa_
**ESI+**	MT-08	3051	368.8	17.7	26.7	4.3	1.1	1.1	1.6	0.3	5.9	−0.7
	MT-15	2432	374.5	18.0	27.4	4.1	1.2	1.1	1.6	0.2	5.9	−0.7
	MT-18	2346	355.6	16.7	24.9	4.1	1.2	1.2	1.6	0.3	5.8	−0.6
	LJ	1481	375.0	19.3	28.7	3.6	1.1	1.0	1.6	0.2	6.5	−0.8
	IS	1691	395.3	18.3	25.0	5.8	1.1	0.8	1.5	0.4	7.4	−0.4
	XJ	2375	363.0	18.0	27.9	4.3	0.9	0.8	1.6	0.3	5.6	−0.8
**ESI−**	MT-08	3741	333.8	16.4	22.0	5.8	0.3	0.6	1.4	0.4	6.6	−0.5
	MT-15	2562	313.2	15.6	21.0	5.2	0.3	0.6	1.4	0.4	6.3	−0.5
	MT-18	2375	304.1	15.1	21.0	5.2	0.2	0.5	1.5	0.4	5.7	−0.6
	LJ	1628	291.0	14.4	20.1	4.8	0.2	0.6	1.5	0.4	5.5	−0.6
	IS	1658	387.9	16.5	22.9	9.5	0.4	0.3	1.6	0.7	6.3	−0.1
	XJ	2783	326.0	16.5	24.9	5.3	0.2	0.5	1.5	0.4	5.2	−0.7

* Intensity-weighted average values are displayed for molecular weight (m/z_wa_), number of carbon (C_wa_), hydrogen (H_wa_), oxygen (O_wa_), nitrogen (N_wa_) and sulfur atoms (S_wa_), hydrogen-to-carbon ratio (H/C_wa_), oxygen-to-carbon ratio (O/C_wa_), double bond equivalent (DBE_wa_) and nominal oxidation state of carbon (NOSC_wa_).

**Table 2 molecules-28-01273-t002:** Number and relative abundance of molecular formulas detected from different samples by ESI+ and ESI− mass spectrometry.

		MT-08	MT-15	MT-18	LJ	IS	XJ
**No.**	ESI+	2193	1851	1751	1217	1442	1774
	ESI−	2883	1981	1780	1364	1389	2182
	Common	858	581	595	264	269	601
**Abund.**	ESI+	61.9%	68.7%	70.6%	81.3%	78.8%	59.9%
	ESI−	66.4%	73.0%	70.2%	85.1%	65.9%	76.3%
	Common	38.1%; 33.6%	31.3%; 27.0%	29.4%; 29.8%	18.7%; 14.9%	21.2%; 34.1%	40.1%; 23.7%

**Table 3 molecules-28-01273-t003:** Specific molecular formulas with the top 10 abundance in different Baijiu samples.

MT-08	MT-15	MT-18	LJ	IS	XJ
C_16_H_26_N_2_OS_2_	C_16_H_26_N_2_OS_2_	C_16_H_26_N_2_OS_2_	C_6_H_12_O_3_	C_15_H_28_O_14_	C_6_H_12_O_3_
C_6_H_12_O_3_	C_15_H_27_N_3_O_5_S_2_	C_14_H_26_N_2_O_2_S_3_	C_15_H_28_N_2_OS_3_	C_12_H_24_O_12_	C_16_H_26_N_2_OS_2_
C_15_H_27_N_3_O_5_S_2_	C_17_H_13_N_3_	C_15_H_27_N_3_O_5_S_2_	C_16_H_26_N_2_OS_2_	C_9_H_18_O_9_	C_15_H_27_N_3_O_5_S_2_
C_9_H_12_N_2_OS	C_6_H_12_O_3_	C_6_H_12_O_3_	C_12_H_10_OS	C_16_H_26_N_2_OS_2_	C_20_H_36_O_5_
C_20_H_38_O_5_	C_14_H_26_N_2_O_2_S_3_	C_9_H_12_N_2_OS	C_17_H_13_N_3_	C_6_H_10_O_5_	C_9_H_12_N_2_OS
C_14_H_26_N_2_O_2_S_3_	C_9_H_12_N_2_OS	C_20_H_38_O_5_	C_13_H_24_N_2_OS_3_	C_23_H_20_O_7_S	C_24_H_46_O_5_
C_19_H_18_O_5_S_3_	C_20_H_38_O_5_	C_17_H_13_N_3_	C_15_H_27_N_3_O_5_S_2_	C_18_H_32_O_16_	C_16_H_24_O_7_
C_17_H_13_N_3_	C_22_H_40_O_4_	C_12_H_22_N_2_O_2_S_3_	C_9_H_12_N_2_OS	C_18_H_34_O_17_	C_17_H_13_N_3_
C_12_H_22_N_2_O_2_S_3_	C_12_H_22_N_2_O_2_S_3_	C_11_H_22_O_6_	C_14_H_26_N_2_O_2_S_3_	C_37_H_36_N_2_O_11_	C_12_H_10_OS
C_13_H_26_N_2_OS_3_	C_12_H_10_OS	C_13_H_26_N_2_OS_3_	C_24_H_24_O_4_	C_9_H_12_N_2_OS	C_14_H_26_N_2_O_2_S_3_

## Data Availability

The authors confirm that the data supporting the findings of this study are available within the article and Appendix A. Raw data that support the findings are available from the corresponding author, upon reasonable request.

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
