# Peer review of "Identification of Trace Components in Sauce-Flavor Baijiu by High-Resolution Mass Spectrometry"

_molecules, 2023, doi:10.3390/molecules28031273_

Round 1

Reviewer 1 Report

This study analyzed the chemical diversity of several representative sauce-flavor Baijiu by the combination of electrospray ionization (ESI) and FT-ICR MS. The results showed that ESI+ and ESI- exhibited different features for the characterization of trace components. However, there are some questions which should be addressed, as listing below:

1. What is the difference between MALDI TOF-MS and FT-ICR MS, and why did you choose FT-ICR MS for the experiment?

2. How many samples are there for each type of sauce-flavor Baijiu? As shown in Figure 1, the samples of each type of sauce-flavor Baijiu are 2, which is too small?

3. The size of the dots represents the abundance of the assigned molecular formulas in figure 4, so what's the exact formula for each dot?

Author Response

Response to Reviewer 1 Comments

This study analyzed the chemical diversity of several representative sauce-flavor Baijiu by the combination of electrospray ionization (ESI) and FT-ICR MS. The results showed that ESI+ and ESI- exhibited different features for the characterization of trace components. However, there are some questions which should be addressed, as listing below:

Point 1: What is the difference between MALDI TOF-MS and FT-ICR MS, and why did you choose FT-ICR MS for the experiment?

Response 1: Yes, thanks for the comments. As mentioned in the manuscript, water and ethanol account for over 98% of components in Baijiu, and the rest trace components are also volatile substances such as esters, alcohols, and aldehydes (page 1, line 41-43). Considering the volatility of Baijiu, many components would be lost during the spotting process of MALDI sample preparation, as well as the fact that the MALDI method is more adapted to the measurement of biological macromolecules, and therefore, the MALDI ion source was not selected in our experiment. In addition, the FT-ICR MS instrument has a much higher mass resolving power compared to TOF-MS. For example, in the following figure, a resolving power over 400,000 is required to distinguish a mass difference of 3.4 milli Daltons in the Baijiu sample here, in order to identify compounds C38H38O12 and C35H42O12S, and such a performance cannot be achieved by TOF-MS. For all the reasons above, to better interpret the complex trace components in Baijiu, we chose ESI and FT-ICR MS for the measurement.

We have also made additional notes in the manuscript: “For sure, aroma is an important attribute affecting the perceived quality of Baijiu, which has a unique collection of volatile chemicals [22]. Further identification of complex mixtures usually requires high-resolution MS, which can be able to resolve multiple peaks with the same nominal m/z [23]. Fourier transform ion cyclotron resonance mass spectrometry (FT-ICR MS) with ultra-high mass resolution has been widely utilized to explore the signature of complex samples at the molecular level, and to reveal elemental formulas (CcHhOoNnSsPp…) of tens of thousands of compounds simultaneously [24-26]. (Page 2, line 77-84)

Point 2: How many samples are there for each type of sauce-flavor Baijiu? As shown in Figure 1, the samples of each type of sauce-flavor Baijiu are 2, which is too small?

Response 2: We apologize for the confusion in the manuscript. Six samples of sauce-flavor Baijiu were analyzed here. They are Maotai manufactured in the year 2008, 2015, and 2018 (MT-08, MT-15, MT-18), respectively, and Langjiu (LJ), Xijiu (XJ), and an Internal Supply brand (IS). Here, we selected representative sauce-flavor Baijiu brands in the market and used ESI+ and ESI- ionization methods respectively to obtain a comprehensive molecular profile of different Baijiu samples. For the reviewer’s interest, the papers listed below also use a smaller number of wine samples to yield solid and meaningful conclusions.

  1. Liu, W.; Zheng, Y.; Zhang, C.; Chen, L.; Zhuang, H.; Yao, G.; Ren, H.; Liu, Y., A biomimetic olfactory recognition system for the discrimination of Chinese liquor aromas. Food Chem. 2022, 386, 132841. (Using 5 samples)
  2. Kew, W.; Mackay, C. L.; Goodall, I.; Clarke, D. J.; Uhrin, D., Complementary Ionization Techniques for the Analysis of Scotch Whisky by High Resolution Mass Spectrometry. Anal Chem 2018, 90, (19), 11265-11272. (Using 4 samples)

To avoid misunderstanding, we have re-expressed the following description in the main text: “Figure 1. Principle component analysis of intensity weighted average values listed in Table 1 for the Baijiu samples measured by ESI+ and ESI-. The red confidence ellipse is the parameter measured by ESI+ mode and the blue confidence ellipse is the parameter measured by ESI- mode. (Page 4, line 136-138)

And we have also provided additional information in the Materials and Methods section: “Four representative types of sauce-flavor Baijiu brands with a total of six samples were selected for mass spectrometric analysis. (Page 9, line 258-259)

Point 3: The size of the dots represents the abundance of the assigned molecular formulas in figure 4, so what's the exact formula for each dot?

Response 3: We are sorry that we didn’t list the detailed molecular formula of each dot in the manuscript. In this revised version, the exact formulas for each dot in Figure 4 were complemented in the supporting material (Supplementary Information). We have also provided additional explanations for the diagram: “The corresponding molecular formulas are listed in details in the supporting material.(Page 6, line 172)

Reviewer 2 Report

The article by Ge et al. regarding Identification of trace components in sauce-flavor Baijiu by high resolution mass spectrometry is well written and within the aims and scopes of Molecules, however the authors need to address the following issues.

The following article should be cited within the introduction as this article is directly related to this article.

Burns, R. L.,. (2021). A Fast, Straightforward and Inexpensive Method for the Authentication of Baijiu Spirit Samples by Fluorescence Spectroscopy. Beverages7(3), 65.

Line 66, a sentence should not be started with “And”.

The materials and methods section needs to come before the results and discussion section. This is a huge problem right now.

In figure 5 some of the numbers at the top of the bar graph are oddly placed.

I would like the authors to explain how they ensured that the instrument was working properly.

Overall this is a well written manuscript, but the authors must address my questions and concerns before I accept this manuscript.

Author Response

Response to Reviewer 2 Comments

The article by Ge et al. regarding Identification of trace components in sauce-flavor Baijiu by high resolution mass spectrometry is well written and within the aims and scopes of Molecules, however the authors need to address the following issues.

Point 1: The following article should be cited within the introduction as this article is directly related to this article.

Burns, R. L.,. (2021). A Fast, Straightforward and Inexpensive Method for the Authentication of Baijiu Spirit Samples by Fluorescence Spectroscopy. Beverages, 7(3), 65.

Response 1: Yes, thanks for the valuable comment. To better illustrate the background of the study, we have read the paper thoroughly and cited it in the manuscript: “In addition, an analytical method based on fluorescence spectroscopy was developed to distinguish liquor products manufactured from six different distilleries in East Asia and North America [13].(Page 2, line 58-60)

Point 2: Line 66, a sentence should not be started with “And”.

Response 2: Thank you for the comment, and we have re-written it: “Fourier transform ion cyclotron resonance mass spectrometry (FT-ICR MS) with ul-tra-high mass resolution has been widely utilized to explore the signature of complex samples at the molecular level, and to reveal elemental formulas (CcHhOoNnSsPp…) of tens of thousands of compounds simultaneously [24-26]. (Page 2, line 80-84)

Point 3: The materials and methods section needs to come before the results and discussion section. This is a huge problem right now.

Response 3: We apologize for the confusion here. However, we have carefully read Instructions for Authors again and found that the current format is in accordance with the journal requirements. Therefore, no changes have been made in the revised version.

Point 4: In figure 5 some of the numbers at the top of the bar graph are oddly placed.

Response 4: Yes, thanks for the comment. We have made changes in Figure 5. (Page 7, line 198)

Point 5: I would like the authors to explain how they ensured that the instrument was working properly.

Response 5: Well, first of all, the temperature in the lab was controlled between 23-24 °C, to keep the instrument's magnetic field, electric field, and pumps working stable. The instrument was quality and quantity controlled by external standard before and after each experiment. During the measurement, a suitable injection speed of 200 μL/h was set, upper and lower threshold values were set for the ion current in the detector. And 256 scans were accumulated for each measurement to obtain one mass spectrum. For characterization, only peaks with a signal-to-noise ratio over 6 were selected for molecular formulas calculation, to ensure accuracy and to avoid interferences.

We have also described the corresponding instrument parameters in the Materials and Methods section:

Baijiu samples were infused at a flow rate of 200 μL/h. For each spectrum, 256 scans were accumulated per run, and the mass range was recorded from 150 to 1000 Da. (Page 9, line 265-267)

In order to minimize the interferences from the noise, only spectral peaks with sig-nal-to-noise ratios above 6 were considered. (Page 9, line 269-271)

Overall this is a well written manuscript, but the authors must address my questions and concerns before I accept this manuscript.

Response: Yes, we are grateful for the reviewer’s valuable comment. We hope that our response addressed all the issues.

Round 2

Reviewer 2 Report

The authors of satisfied my requests. This manuscript is ready for publication.